# Evaluation of *Staphylococcus aureus* Colonization in Adult Patients Undergoing Tonsillectomy for Recurrent Tonsillitis

**DOI:** 10.3390/pathogens11040427

**Published:** 2022-03-31

**Authors:** Renata Klagisa, Karlis Racenis, Renars Broks, Ligija Kise, Juta Kroiča

**Affiliations:** 1Department of Otorhinolaryngology, Daugavpils Regional Hospital, LV-5401 Daugavpils, Latvia; 2Department of Doctoral Studies, Riga Stradins University, LV-1007 Riga, Latvia; profesorekise@gmail.com; 3Department of Biology and Microbiology, Riga Stradins University, LV-1007 Riga, Latvia; karlis.racenis@rsu.lv (K.R.); renars.broks@rsu.lv (R.B.); juta.kroica@rsu.lv (J.K.); 4Center of Nephrology, Pauls Stradins Clinical University Hospital, LV-1002 Riga, Latvia

**Keywords:** biofilm, carrier, colonization, recurrent tonsillitis, *Staphylococcus aureus*

## Abstract

Background and objectives: *Staphylococcus aureus* (*S. aureus*) is often recovered from the pharynx. However, the role of this pathogen in the etiology of tonsillar inflammation is still unclear and complicated due to frequent carriage of *S. aureus*. The aim of the study was to evaluate the frequency and the clinical importance of *S. aureus* colonization and biofilm production ability in patients with recurrent tonsillitis (RT) using patient samples from tonsillar crypts during tonsillectomy, and from the throat, nasal cavity, and armpits after tonsillectomy. Materials and Methods: A case series study was carried out at a tertiary referral center among 16 patients diagnosed with RT who were undergoing tonsillectomy. Samples from tonsillar crypts were obtained during tonsillectomy, and samples from the throat, nasal cavity, and armpit were obtained a year after surgery. An evaluation of *S. aureus* incidence, biofilm formation, and antibacterial susceptibility was performed. Results: During tonsillectomy, 16 strains of *S. aureus* were isolated from 16 patients, while 15/16 *S. aureus* strains were biofilm producers. A year after tonsillectomy, 8 *S. aureus* strains were isolated from 6 out of 16 patients, while 6/8 *S. aureus* strains were biofilm producers. After tonsillectomy, 3 patients showed *S. aureus* in throat culture. Conclusions: In 10/16 cases *S. aureus* was the causative agent of RT, in 4/16 cases patients had a predisposition to colonization of *S. aureus*, and in 2/16 cases *S. aureus* was a part of the patients` oral microbiome. Tonsillectomy results in less frequent isolation of *S. aureus* strains.

## 1. Introduction

The ecological niche of *Staphylococcus aureus* (*S. aureus*) in humans is the anterior nares, however it can be frequently isolated from the throat, palatine tonsils, and skin [1]. Approximately 20–25% of the healthy adult population have become persistently colonized by *S. aureus*, 60% intermittently, while 20–30% of population are non-carriers [2]. Hanson et al. (2018) reported that in the USA, from 177 adults, 6.2% carried *S. aureus* only in the anterior nares, 18.6% only in the oropharynx, and 19.8% in both sites [3]. They found out that the prevalence of oropharyngeal carriage was higher in urban (47.3%) than rural (27.9%) environments [3]. Chmielowiec-Korzeniowska et al.’s (2020) study demonstrated that every third adult (32%) in Poland was an asymptomatic carrier of *S. aureus*, and *S. aureus* was recovered from the pharynx more often than from the nose or skin [2].

Previous studies have shown that there is a strong causal connection between *S. aureus* nasal carriage and increased risk of nosocomial infection in dialysis patients and in those undergoing surgery [1,4]. *S. aureus* is an etiological factor of such purulent infections as furuncles, abscesses, osteomyelitis, and sepsis. However, the role of this pathogen in the etiology of tonsillar inflammation is still unclear and complicated due to frequent carriage of *S. aureus*. Aside from *S. aureus*, other aerobic and anaerobic pathogens are also implicated in tonsil infections, for example alpha and beta hemolytic Streptococcus (group A, C), *Hemophilus influenzae*, *Haemophilus parainfluenzae*, *Enterococcus* spp., *Klebsiella pneumoniae*, *Corynebacterium* spp., *Peptostreptococci*, Fusobacteria, *Bacteroides,* and *Veillonella* [5,6,7]. Oral microbiota, which are potential pathogens, may make it difficult to identify the causative agent of tonsillitis. Researchers claim that routine culture of surface swab specimens in patients with recurrent tonsillitis (RT) is not reliable and recommend core sampling as the diagnostic method of choice [6,8,9].

Regardless of the widespread use of antibiotics to treat infected tonsils, episodes of tonsillitis tend to recur and form a condition known as recurrent tonsillitis (RT). Episodes of tonsillitis decrease quality of life and are a financial burden due to absences in school or work and health care costs [6]. RT can be treated with tonsillectomy in those patients in whom at least five or more attack episodes occur in a year [5,10]. If less than three episodes are observed up to the time of first presentation, the indication for surgery cannot be made until at least six episodes occur within the observation period [10].

Importantly, *S. aureus* strains can acquire broad antibacterial resistance. Parts of *S. aureus* strains isolated from patients with RT were considered multidrug-resistant and methicillin-resistant (MRSA) [11,12]. In the Netherlands, healthcare workers who are MRSA carriers achieved successful MRSA eradication only after tonsillectomy [13]. Antibiotic resistance can be explained by inadequate penetration of antibiotics into the tonsillar core, the protection of bacteria within epithelial cells and macrophage-like cells, the resistance of strains to the typical antibiotic treatments due to repeated antibiotic courses, and the prevalence of biofilm-producing bacteria (5,6). According to Brook and Foote’s (2006) study, *S. aureus* strains are more often found in the tonsillar core than on the tonsillar surface [14]. *S. aureus* persistence in tonsillar tissues is still a matter of discussion and requires further research [15].

The ability of *S. aureus* to produce multilayered, mature biofilms may contribute to the survival of *S. aureus* in tonsillar tissue and play an important role in the persistence of chronic infection [16,17,18]. Growth in biofilm provides a defense against host immune responses, can impede the access of macrophages, and can increase the tolerance to antibiotics [16,17]. Biofilm-associated antibiotic tolerance is a transient state in which normally susceptible bacteria enter homeostasis, which decreases sensitivity [19]. When these cells disperse and re-enter a plankton state, they regain normal antibiotic sensitivity [19]. In this study, we investigated the biofilm formation ability of *S. aureus* compared to biofilm production among strains and used it as a screening method to evaluate the clinical role of *S. aureus* in each study patient with RT. Samples from tonsillar crypts were obtained with punch biopsy needles, which is a novel technique for tonsillar sampling.

The objectives here were to evaluate the frequency and clinical importance of *S. aureus* colonization and biofilm production ability in patients with RT using patient samples from tonsillar crypts during tonsillectomy, and from the throat, nasal cavity, and armpits after tonsillectomy.

## 2. Results

The study group included 8 females and 8 males aged between 21 and 50 years, with a mean age of 29 years (±7.23). Seven patients lived in Riga, with the rest living in other cities in Latvia (Table 1).

During the past 3 years, 16 patients had from 2 to 7 recurrent episodes of tonsillitis. Three patients had a peritonsillar abscess in their medical history. Two patients had cryptolysis with radiofrequency, while 3 patients had cryotherapy before surgery. The last antibacterial treatment was received no earlier than 1 month before surgery. Three patients were hospitalized (1 patient in January 2018, 2 patients in 2017).

From tonsillar crypts the most commonly isolated bacteria was *S. aureus*, being the only microorganism in 6 patients and co-isolated with oral flora or with other potentially pathogenic microorganisms in 10 patients. Regarding *S. aureus*, 16 strains were isolated and tested for biofilm production, with 15/16 strains being biofilm producers. Furthermore, 1 of the strains was a strong biofilm producer, 5/16 strains were moderate, 9/16 were weak, and 1 strain of *S. aureus* did not produce a biofilm (Table 1).

One year after tonsillectomy, 8 *S. aureus* strains were isolated from 6 out of 16 patients—from throat cultures in 3/16 patients, from nasal samples in 4/16 patients, and from armpit samples in 1/16 patients. From throat samples, 1/3 was a strong biofilm producer and 2/3 strains of *S. aureus* were weak biofilm producers. From nasal samples, 1/4 was a moderate biofilm producer, 1/4 was a weak biofilm producer, and 2/4 strains of *S. aureus* did not produce a biofilm. From armpit samples, 1 strain of *S. aureus* did not produce biofilm. Only one patient had *S. aureus* in the throat, nasal, and armpit samples, and was a weak biofilm producer (Table 1).

*S. aureus* strains were isolated in 10 patients only during tonsillectomy; in these cases *S. aureus* was a causative agent of RT. Four patients had *S. aureus* in the throat or nasal cavity after tonsillectomy, but *S. aureus* isolates showed different degrees of biofilm formation. In these cases, patients had predisposition to *S. aureus* colonization. In 2 patients, *S. aureus* with the same biofilm production capacity was isolated from palatine tonsils during tonsillectomy and 1 year after tonsillectomy from the throat, nasal cavity, and axilla. In these cases, *S. aureus* was a part of the patients’ microbiome (Table 1).

Figure 1 illustrates the biofilm formation of 11 *S. aureus* isolates in a 96-well flat-bottom microtiter plate. The crystal violet absorption is proportional to the adhesion cells and concentration of biofilm.

The crystal violet dye attached to the cells forming biofilms on microtiter plates was quantified. The optical density (OD) of the bacterial biofilm was measured at 570 nm wavelength with a microplate spectrophotometer. The OD values for each strain were expressed as a number. All mean OD values of isolated strain biofilms are summarized in Figure 2.

Fourteen isolates were resistant to benzylpenicillin and ampicillin. One isolate was identified as MRSA, which was resistant to benzylpenicillin, ampicillin, cefoxitin, ceftriaxone, ampicillin–sulbactam, and amoxicillin with clavulanic acid, but intermediate resistance to ciprofloxacin. All isolates showed intermediate resistance to ciprofloxacin, while 1 isolate also showed intermediate resistance to clindamycin (Table 2).

## 3. Discussion

The anterior nares represent the dominant ecological niche, while other sites that can be colonized include the axilla, perineum, and pharynx [4]. Elimination of the nasal carriage by topical antibiotics generally leads to loss of carriage in these areas [20]. *S. aureus* readily recolonizes the nose, throat, and other sites within several months after antibiotic treatment [21,22]. In *S. aureus* carriers, infection rates are higher than in non-carriers, and patients are usually infected by the same strains with which they are colonized [17]. In our study, infected palatine tonsils were removed surgically, and the presence of *S. aureus* in the pharynx, axilla, and nares was assessed 1 year after tonsillectomy.

There are host and bacterial factors that can influence the carriage of *S. aureus*. The main predisposing factors to staphylococcal infection development include age, the presence of chronic diseases or immunodeficiency, genetics, direct contact with healthcare settings, and hospitalization [2]. Patients enrolled in the study were young individuals (mean age 29 years) without high carriage rate host factors such as HIV infection, insulin-dependent diabetes, continuous ambulatory peritoneal dialysis and hemodialysis, or intravenous drug use [4,23,24,25]. The mean age of our study patients was consistent with the data from other studies, in which tonsillectomy patients were 28 years old [26,27].

We are aware of the disadvantages of routine surface sampling of tonsils. Tonsillar core and tonsillar crypt samples are more favorable for culturing [6,8]. Crypts are narrow passages that penetrate the palatine tonsils. It is not possible to obtain the contents of crypts with a cotton swab. In our study, samples from the palatine tonsils were obtained with a punch biopsy needle, which is suitable for the width and depth of the crypts to obtain optimal size samples [28].

Biofilm formation is one of the bacterial factors that is distinctive for the adhesive phenotype of bacteria [4]. Bacteria in biofilm state present differential metabolic and physiological functions, often rendering them more virulent and resistant to antibiotics [29]. Neopane and co-authors (2018) showed that biofilm-producing *S. aureus* isolated from wounds was more resistant to various antimicrobials than the biofilm non-producers [30].

A broad range of assays for biofilm quantification in microtiter plates have been described [31,32]. We used the crystal violet assay adapted from Stepanovic et al. (2007) because it is reliable, cost-effective, straightforward, and is commonly used for the quantification of biofilm production by staphylococci [33]. It is also important that this method can be easily performed by other investigators. Because both living and dead cells, as well as the matrix, are stained by the crystal violet dye, this method is poorly suited for differentiation between living and dead cells, and susceptibility testing of biofilms cannot be performed [31]. The drawbacks of the method do not affect the research question being investigated in our study. However, other methods such as flow cell systems would increase data reliability and would allow more detailed biofilm investigations, specifically in dynamic conditions.

In our study, emphasis was placed on the biofilm formation ability of *S. aureus* isolated strains. *S. aureus* strains isolated from the tonsillar crypts, as compared to isolates collected from other body sites had greater capacity to produce biofilms. Biofilm-producing *S. aureus* strains were mostly isolated from tonsillar crypts and were susceptible to the majority of tested antibiotics. Only one isolate was identified as MRSA, which showed a wider spectrum of resistance and was a weak biofilm producer. The isolate was obtained from a 25-year-old female without co-morbidities, with 5 episodes of tonsillitis per year for the last 3 years. After tonsillectomy the MRSA strain was eradicated. MRSA strains were detected in RT patients in other studies also. In the study conducted by Katkowska et al., the MRSA strain was isolated from the tonsils in one out of 118 adult patients and in two out of 73 children qualified for tonsillectomy in Poland [7,12]. The role of biofilm formation and the antimicrobial resistance of MRSA and methicillin-susceptible *S. aureus* (MSSA) are unclear [34]. The environmental factors (temperature, pH, glucose level, type of media, and others) influence bacterial biofilm production. Therefore, these factors should be accounted for in biofilm research. To compare the results from different studies, one should use similar or even the same biofilm method and environmental factors [35]. Our study results showed that in 10 patients tonsillectomy resulted in no growth of *S. aureus* strains a year later; therefore, tonsillectomy could prevent bacterial colonization within a one year period. However, to prove such phenomena, this should be investigated in bigger cohorts.

Our study underlines the immense importance of studying *S. aureus* colonization to understand the pathology of staphylococcal disease. Current efforts to interrupt carriage rely on the use of antibiotics, but the development of efficacious antibiofilm *S. aureus* therapies is a new and necessary perspective.

## 4. Strengths and Limitations

The study provides a one year follow-up period for surgically treated patients with recurrent tonsillitis. Furthermore, this study provides evaluations of the outcomes and the effects of tonsillectomy on the results of microbiological testing.

However, several limitations should be mentioned. Firstly, the small number of cases observed during the study period could cause bias. Further studies, including those with a larger study group, a control group, an increased bacterial spectrum with biofilm formation, antibacterial susceptibility, and bacterial genotyping, would be necessary to draw reliable conclusions regarding tonsillitis.

## 5. Materials and Methods

A case series study was performed in Pauls Stradins Clinical University Hospital in Riga, Latvia. The study lasted from 2018 to 2020. The study was approved by the local ethics committee of Riga Stradins University (document No. 49/30 November 2017). Written consent was obtained from every patient. The exclusion criterion was recent (less than a month ago) antibacterial treatment. A control group was not applied. Patient medical history information was gathered from patients and via their medical records by a general practitioner and anesthesiologist before elective tonsillectomy.

### 5.1. Isolation of Microorganisms and Antibacterial Susceptibility Testing

Samples from tonsillar crypts were obtained with a punch biopsy needle from 16 adults undergoing tonsillectomy for RT. For research purposes, the punch biopsy needle was designed with a prolonged and curved handle and a circular blade for tonsillar crypt biopsy (patent number: LVP2020000055) [28]. Swab samples from the throat, nasal cavity, and armpit were obtained a year later to assess *S. aureus* carriage. Materials were transported with universal transport media (AMIES) at room temperature within 24–48 h.

The obtained materials were inoculated in blood agar, chocolate agar (with test discs loaded with oleandomycin in a CO_2_-enriched atmosphere), CAN agar (with optochin disk test), Brucella blood agar, mannitol salt agar, MacConkey agar, and Saburo agar plates and incubated under aerobic conditions for 48 h at 36 ± 2 °C temperature. The isolated microorganisms underwent the macro- and microscopic evaluations. The isolated bacteria were identified by Gram staining using a VITEK-2 Compact device (bioMériux, Marcy l’Étoile, France).

Disk diffusion antimicrobial susceptibility tests were performed and the results were evaluated according to the recommendations of the European Committee on Antimicrobial Susceptibility Testing (EUCAST), *Clinical Breakpoints and Dosing of Antibiotics* (Version 10.0, January 2020) [36].

### 5.2. Biofilm Growth Using Cristal Violet Assay

A crystal violet assay adapted from Stepanovic et al. (2007) was used for the in vitro cultivation and quantification of bacterial biofilms [33]. Isolated *S. aureus* strains were suspended in trypticase soy broth (TSB) supplemented with an additional 1% glucose for incubation at 37 ℃ for 16–18 h. Inoculated broths were diluted with un-inoculated TSB at a ratio of 1:100. Then, 150 µL of diluted suspension was transferred using a multichannel pipette into a sterile 96-well plate (Thermo Scientific™ Nunc MicroWell 96-Well Microplates (flat-bottomed), Thermo Fisher Scientific, Roskilde, Denmark). Each plate contained 11 strains and the negative control (sterile broth) at 8 wells per strain, and each experiment was performed in triplicate. The inoculated plates were cultivated aerobically at 37 °C for 48 h. After incubation, all wells were emptied by throwing out the liquid in a clinical waste bag without using a pipette. Each well was rinsed 3 times with 250 µL 0.9% saline. After washing, staining was performed by adding 150 µL of 0.1% crystal violet per well. After 15 minutes, the color was removed by throwing it out and each well was washed 3 times with 250 µL distilled water. At the end, 150 µL of 96% ethanol was added to each well. Afterwards, the optical density (OD) of wells was measured at 570 nm wavelength with a microplate spectrophotometer (Tecan Infinite F50, Mannedorf, Switzerland, with Magellan™ reader control and data analysis software V 6.6) [37].

### 5.3. Biofilm Calculation

The OD values for each strain were averaged and expressed as a number. The cut-off value (ODc) was defined as three standard deviations above the mean OD of the negative control and was separately calculated for each experiment. Strains were divided as follows: OD ≤ ODc = no biofilm producer, ODc < OD ≤ 2 × ODc = weak biofilm producer, 2 × ODc < OD ≤ 4 × ODc = moderate biofilm producer, 4 × ODc < OD = strong biofilm producer [25].

### 5.4. Data Analysis

Data analysis was performed using SPSS software (IBM SPSS Statistics version 26) and Microsoft Excel 10.5.5. Assessment of Outcomes

Outcomes were categorized as:(1)A causative agent of RT—*S. aureus* strains were isolated only in tonsillar crypts, while no *S. aureus* was recovered from any site after tonsillectomy;(2)Predisposition to colonization—*S. aureus* strains were isolated during and also after tonsillectomy, but *S. aureus* strains from one individual showed different phenotypes in their biofilm formation profiles;(3)Parts of patients’ oral microbiomes—*S. aureus* strains were isolated during and also after tonsillectomy, but *S. aureus* strains showed no phenotypical changes in biofilm formation.

## 6. Conclusions

*S. aureus* is a common cause of recurrent tonsillitis in our cohort population. Tonsillectomy results in less frequent isolation of *S. aureus* strains. Isolation of *S. aureus* from multiple sites and the determination of the biofilm production capacity of isolated strains is a way of distinguishing carriers of *S. aureus*.

## 7. Patents

Klagisa, R.; Kroica, J.; Kise, L. Punch Biopsy Needle. Patent No: LVP2020000055. In *Izgudrojumi, Preču Zīmes un Dizain-paraugi*. Patent Office of the Republic of Latvia, Riga, Latvia, 2021; Volume 5, pp. 315.

## Figures and Tables

**Figure 1 pathogens-11-00427-f001:**
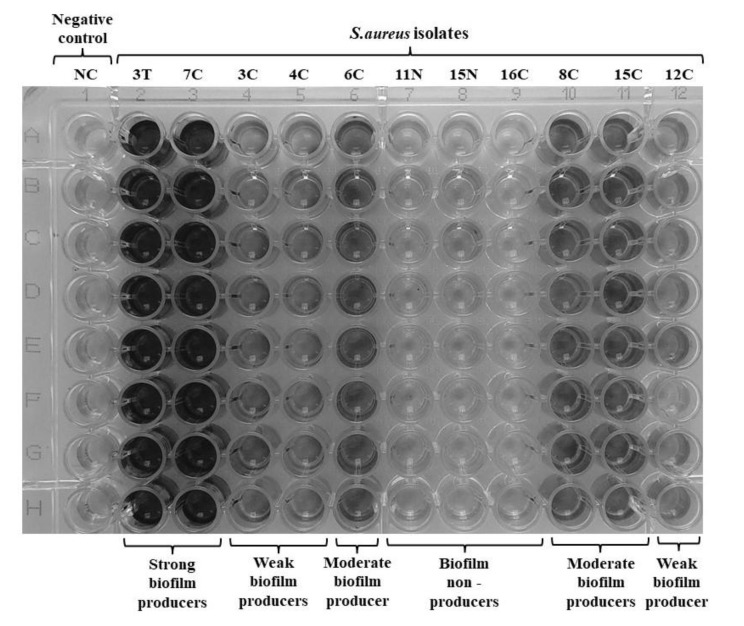
The biofilm formation of *S. aureus* isolates in a 96-well flat-bottom microtiter plate. The plate contained 11 strains and the negative control (NC) at 8 wells per strain. Staining was performed with crystal violet dye, differentiating strong (3T, 7C), moderate (6C, 8C, 15C), and weak biofilm producers (3C, 4C, 12C) and biofilm non-producers (11N, 15N, 16C). *S.aureus* isolate code – the number designates the patient and the letter - the carriage site (C—tonsillar crypts; T—throat; N—nasal cavity).

**Figure 2 pathogens-11-00427-f002:**
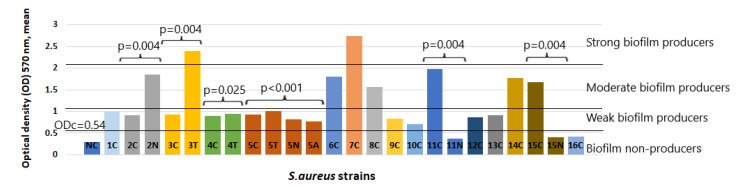
Biofilm production capability on microtiter plates of 24 isolates of *S. aureus*. Bars represent mean values of OD (measured at a wavelength of 570 nm). Trypticase soy broth with 1% glucose as a negative control (NC). The cut-off value (ODc) and biofilm production capacity levels are marked with horizontal lines. Differences between suspect *S. aureus* strain capability in biofilm formation were analyzed using Mann–Whitney U test for two-strain analysis and Kruskal-Wallis test for four-strain analysis, and are expressed as *p*-values. *S.aureus* strain code–the number designates the patient and the letter - the carriage site (C—tonsillar crypts; T—throat; N—nasal cavity; A—armpits).

**Table 1 pathogens-11-00427-t001:** Patient characteristics and results of microbiological testing during and one year after tonsillectomy.

N	Sex	Age	Place of Residence	Microorganisms	*S. aureus* Biofilm Production				
					During TE	After TE			
					Tonsillar Crypts	Throat	Nasal Cavity	Armpits	
1	M	35	Bauska	***S. aureus*** + oral flora	1C weak	0	0	0	**Causative agent**
2	M	21	Saulkrasti	** *S. aureus* **	2C weak	0	2N moderate	0	**Predisposition to colonization**
3	F	27	Jurmala	***S. aureus*** + *Candida* spp.	3C weak	3T strong	0	0	**Predisposition to colonization**
4	F	25	Riga	***S. aureus*** + oral flora	4C weak	4T weak	0	0	**Part of patients’ oral microbiome**
5	M	23	Babite	***S. aureus*** + oral flora	5C weak	5T weak	5N weak	5A weak	**Part of patients’ oral microbiome**
6	M	50	Riga	** *S. aureus* ** *+ Staphylococcus epidermidis*	6C moderate	0	0	0	**Causative agent**
7	F	35	Marupe	***S. aureus*** + oral flora	7C strong	0	0	0	**Causative agent**
8	M	24	Salaspils	** *S. aureus* **	8C moderate	0	0	0	**Causative agent**
9	F	33	Riga	***S. aureus*** + *Streptococcus pneumoniae + oral flora*	9C weak	0	0	0	**Causative agent**
10	F	31	Riga	***S. aureus*** + *Klebsiella pneumoniae* + *Candida* spp. + oral flora	10C weak	0	0	0	**Causative agent**
11	M	35	Riga	** *S. aureus* **	11C moderate	0	11N non—producer	0	**Predisposition to colonization**
12	M	24	Riga	** *S. aureus* **	12C weak	0	0	0	**Causative agent**
13	M	33	Saulkrasti	***S. aureus*** + *Neisseria subflava* + *Haemophilus influenzae* + *Streptococcus anginosus* + *Prevotella intermedia + oral flora*	13C weak	0	0	0	**Causative agent**
14	F	23	Ozolnieki	** *S. aureus* **	14C moderate	0	0	0	**Causative agent**
15	F	31	Rezekne	***S. aureus*** + oral flora + *Streptococcus agalactiae*	15C moderate	0	15N non—producer	0	**Predisposition to colonization**
16	F	28	Riga	** *S. aureus* **	16C non—producer	0	0	0	**Causative agent**

Note: M—male; F—female; TE—tonsillectomy; C—tonsillar crypts; T—throat; N—nasal cavity; A—armpits.

**Table 2 pathogens-11-00427-t002:** Antibiotic resistance among *S. aureus* strains isolated from patients with RT.

		Antibiotics										
		FOX	CRO	P	AMP	AMS	AUG	CIP	AK	E	CD	C
** *S.aureus* ** **strains**	1C	S	S	R	R	S	S	I	S	S	S	S
	2C	S	S	S	S	S	S	I	S	S	S	S
	2N	S	S	R	R	S	S	I	S	S	S	S
	3C	S	S	R	R	S	S	I	S	S	S	S
	3T	S	S	S	S	S	S	I	S	S	S	S
	4C	R	R	R	R	R	R	I	S	S	S	S
	4T	S	S	R	R	S	S	I	S	S	S	S
	5C	S	S	R	R	S	S	I	S	S	S	S
	5T	S	S	S	S	S	S	I	S	S	S	S
	5N	S	S	S	S	S	S	I	S	S	S	S
	5A	S	S	S	S	S	S	I	S	S	S	S
	6C	S	S	R	R	S	S	I	S	S	S	S
	7C	S	S	R	R	S	S	I	S	S	S	S
	8C	S	S	R	R	S	S	I	S	S	S	S
	9C	S	S	R	R	S	S	I	S	S	S	S
	10C	S	S	R	R	S	S	I	S	S	S	S
	11C	S	S	S	S	S	S	I	S	S	S	S
	11N	S	S	S	S	S	S	I	S	S	S	S
	12C	S	S	R	R	S	S	I	S	S	S	S
	13C	S	S	R	R	S	S	I	S	S	I	S
	14C	S	S	S	S	S	S	I	S	S	S	S
	15C	S	S	R	R	S	S	I	S	S	S	S
	15N	S	S	R	R	S	S	I	S	S	S	S
	16C	S	S	S	S	S	S	I	S	S	S	S

FOX, cefoxitin; CRO, ceftriaxone; P, benzylpenicillin; AMP, ampicillin; AMS, ampicillin–sulbactam; AUG, amoxicillin–clavulanic acid; CIP, ciprofloxacin; AK, amikacin; E, erythromycin; CD, clindamycin; C, chloramphenicol. S, sensitive; R, resistant; I, intermediate. *S.aureus* strain code–the number designates the patient and the letter-the carriage site (C—tonsillar crypts; T—throat; N—nasal cavity; A—armpits). Coloring: *S.aureus* strain code colors correspond to the optical density bars of the same isolates in Figure 2; greyish–antibiotics; rosy–resistant; yellowish–intermediate.

## Data Availability

The datasets generated are available from the corresponding author upon reasonable request.

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
