# Peer review of "Evaluation of Staphylococcus aureus Colonization in Adult Patients Undergoing Tonsillectomy for Recurrent Tonsillitis"

_pathogens, 2022, doi:10.3390/pathogens11040427_

Round 1
Reviewer 1 Report
The aim of this study was to evaluate the frequency and the clinical importance of Staphylococcus aureus colonization and biofilm production ability in patients with recurrent tonsillitis (RT). In the first stage of the research, samples were taken from the tonsil crypts during tonsillectomy. One year after tonsillectomy, throat, nasal and armpit swabs were collected from the same patients to assess S. aureus carrier status.
Doubts may be raised by the small number of cases observed during the study period.
The manuscript is well organized.
In my opinion, the article presented for review, contains valuable information but please read the comments.
Please explain and introduce corrections.
- In verse 87-89 it says that: „Twelve isolates were resistant to benzylpenicillin and ampicillin, 1 isolate was resistant to benzylpenicillin, ampicillin, cefoxitin, ceftriaxone, ampicillin – sulbactam and amoxicillin with clavulanic acid, but intermediate resistant to ciprofloxacin.”, and in verse 90-91: “None of the isolated strains was methicillin resistant”.
If 1 strain was resistant to cefoxitin, why was no strain resistant to methicillin? According to EUCAST, cefoxitin is an indicator of methicillin resistance.
- In verse 211 it says: „5. Conclusions”. Should be: 6. Conclusions
- Literature from the last 5 years accounts for only 35%. Please explain why?
Author Response
Thank you for valuable and constructive criticism. We addressed Your comments and suggestions point by point. Changes have been marked by green color.
- In verse 87-89 it says that: „Twelve isolates were resistant to benzylpenicillin and ampicillin, 1 isolate was resistant to benzylpenicillin, ampicillin, cefoxitin, ceftriaxone, ampicillin – sulbactam and amoxicillin with clavulanic acid, but intermediate resistant to ciprofloxacin.”, and in verse 90-91: “None of the isolated strains was methicillin resistant”.
If 1 strain was resistant to cefoxitin, why was no strain resistant to methicillin? According to EUCAST, cefoxitin is an indicator of methicillin resistance.
Response: Thank you, this is corrected now.
One isolate was identified as MRSA, it was resistant to benzylpenicillin, ampicillin, cefoxitin, ceftriaxone, ampicillin – sulbactam and amoxicillin with clavulanic acid, but intermediate resistant to ciprofloxacin.
- In verse 211 it says: „5. Conclusions”. Should be: 6. Conclusions
Response: Thank you, this is corrected now.
- Conclusions
- Literature from the last 5 years accounts for only 35%. Please explain why?
Response: There are not many studies investigating the role of S.aureus in the pathogenesis of tonsillitis. Therefore, articles from a longer period were taken into account. Per your recommendations the Literature was reviewed. The additional latest literature sources were used and added to the References.
References
- Mertz, D.; Frei, R.; Jaussi, B.; Tietz, A.; Stebler, C.; Fluckiger, U.; Widmer, F.A. Throat Swabs Are Necessary to Reliably Detect Carriers of Staphylococcus aureus. Clin Infect Dis. 2007, 45, 475–7. https://doi.org/10.1086/520016
- Chmielowiec-Korzeniowska, A.; Tymczyna, L.; Wlazło, Ł.; Nowakowicz-Dębek, B.; Trawińska, B. Staphylococcus aureus carriage state in healthy adult population and phenotypic and genotypic properties of isolated strains. Postepy Dermatol Alergol. 2020, 37, 184–9. doi: 10.5114/ada.2020.94837
- Hanson, B.M.; Kates, A.E.; O’Malley, S.M.; Mills, E.; Herwaldt, L.A.; Torner, J.C.; Dawson, J.D.; Farina, S.A.; Klostermann, C.; Wu, J.Y.; Quick, M.K.; Forshey, B.M.; Smith, T.C. Staphylococcus aureus in the nose and throat of Iowan families. Epidemiol Infect. 2018, 146, 1777–84. doi: 10.1017/S0950268818001644
- Peacock, S.J.; de Silva, I.; Lowy, F.D. What determines nasal carriage of Staphylococcus aureus? Trends Microbiol. 2001 9, 605–10. DOI: 10.1016/s0966-842x(01)02254-5
- Buname, G.; Kiwale, G.A.; Mushi, M.F.; Silago, V.; Rambau, P.; Mshana, S.E. Bacteria Patterns on Tonsillar Surface and Tonsillar Core Tissue among Patients Scheduled for Tonsillectomy at Bugando Medical Centre, Mwanza, Tanzania. Pathogens. 2021, 10, 1560. https://doi.org/10.3390/pathogens10121560
- Dickinson, A.; Kankaanpää, H.; Silén, S.; Meri, S.; Haapaniemi, A.; Ylikoski, J.; Mäkitie, A. Tonsillar surface swab bacterial culture results differ from those of the tonsillar core in recurrent tonsillitis. The Laryngoscope. 2020, 130, E791–E794. https://doi.org/10.1002/lary.28403
- Katkowska, M.; Garbacz, K.; Kopala, W.; Schubert, J.; Bania, J. Genetic diversity and antimicrobial resistance of Staphylococcus aureus from recurrent tonsillitis in children. APMIS. 2020, 128, 211–9. DOI: 10.1111/apm.13007
- Windfuhr, J.P.; Toepfner, N.; Steffen, G.; Waldfahrer, F.; Berner, R. Clinical practice guideline: tonsillitis II. Surgical management. Eur Arch Otorhinolaryngol. 2016, 273, 989–1009. DOI: 10.1007/s00405-016-3904-x
- Cavalcanti, V.P.; Camargo, L.A. de; Moura, F.S.; Fernandes, E.J. de M.; Lamaro-Cardoso, J.; Braga, C.A. da S.B.; André, M.C.P. Staphylococcus aureus in tonsils of patients with recurrent tonsillitis: prevalence, susceptibility profile, and genotypic characterization. Braz J Infect Dis. 2019, 23, 8–14. DOI: 10.1016/j.bjid.2018.12.003
- Katkowska, M.; Garbacz, K.; Stromkowski, J. Staphylococcus aureus isolated from tonsillectomized adult patients with recurrent tonsillitis. 2017, 125, 46–51. DOI: 10.1111/apm.12628
- Labordus-van Helvoirt R.E.M.; van Rijen, M.M.L.; van Wijngaarden, P. Tonsillectomy for persistent MRSA carriage in the throat—Description of three cases. Int J Infect Dis. 2018, 67, 98–101. DOI: 10.1016/j.ijid.2017.12.007
- Brook, I.; Foote, P.A. Isolation of methicillin resistant Staphylococcus aureus from the surface and core of tonsils in children. Int J Pediatr Otorhinolaryngol. 2006, 70, 2099–102. DOI: 10.1016/j.ijporl.2006.08.004
- Zautner, A.E.; Krause, M.; Stropahl, G.; Holtfreter, S.; Frickmann, H.; Maletzki C.; Kreikemeyer, B.; Pau, W.H.; Podbielski, A. Intracellular Persisting Staphylococcus aureus Is the Major Pathogen in Recurrent Tonsillitis. PLoS ONE. 2010, 5, e9452. DOI: 10.1371/journal.pone.0009452
- Archer, N.K.; Mazaitis, M.J.; Costerton, J.W.; Leid, J.G.; Powers, M.E.; Shirtliff, M.E. Staphylococcus aureus biofilms. 2011, 2, 445–59. DOI: 10.4161/viru.2.5.17724
- Lister, J.L.; Horswill, A.R. Staphylococcus aureus biofilms: recent developments in biofilm dispersal. Front Cell Infect Microbiol. 2014, 4, 178. https://doi.org/10.3389/fcimb.2014.00178
- Moormeier, D.E.; Bayles, K.W. Staphylococcus aureus Biofilm: A Complex Developmental Organism. Mol Microbiol. 2017, 104, 365–76. DOI: 10.1111/mmi.13634
- Singh, R.; Ray, P.; Das, A.; Sharma, M. Role of persisters and small-colony variants in antibiotic resistance of planktonic and biofilm-associated Staphylococcus aureus: an in vitro study. J Med Microbiol. 2009, 58, 1067–73. DOI: 10.1099/jmm.0.009720-0
- Reagan, R.D.; Doebbeling,N.B.; Pfaller,A.M.; Sheetz,T.C.; Houston, K.A.; Hollis, J.R.; Wenzel, P.R. Elimination of Coincident Staphylococcus aureus Nasal and Hand Carriage with Intranasal Application of Mupirocin Calcium Ointment. Annals of Internal Medicine. 1991, 144, 101-106.
- Mody, L.; Kauffman, C.A.; McNeil, S.A.; Galecki, A.T.; Bradley, S.F. Mupirocin-Based Decolonization of Staphylococcus aureus Carriers in Residents of 2 Long-Term Care Facilities: A Randomized, Double-Blind, Placebo-Controlled Trial. Clin Infect Dis. 2003, 37, 1467–74. DOI: 10.1086/379325
- Coates, T.; Bax, R.; Coates, A. Nasal decolonization of Staphylococcus aureus with mupirocin: strengths, weaknesses and future prospects. J Antimicrob Chemother. 2009, 64, 9–15. DOI: 10.1093/jac/dkp159
- Brandwein, M.; Steinberg, D.; Meshner, S. Microbial biofilms and the human skin microbiome. Npj Biofilms Microbiomes. 2016, 2, 3. https://doi.org/10.1038/s41522-016-0004-z
- Neopane, P.; Nepal, H.P.; Shrestha, R.; Uehara, O.; Abiko, Y. In vitro biofilm formation by Staphylococcus aureus isolated from wounds of hospital-admitted patients and their association with antimicrobial resistance. Int J Gen Med. 2018, 11, 25–32. DOI: 10.2147/IJGM.S153268
- Peeters, E.; Nelis, H.J.; Coenye, T. Comparison of multiple methods for quantification of microbial biofilms grown in microtiter plates. J Microbiol Methods. 2008, 72, 157–65. DOI: 10.1016/j.mimet.2007.11.010
- Grossman, A.B.; Burgin, D.J.; Rice, K.C. Quantification of Staphylococcus aureus Biofilm Formation by Crystal Violet and Confocal Microscopy. Methods Mol Biol. 2021, 2341, 69-78. doi:10.1007/978-1-0716-1550-8_9
- Stepanović, S.; Vuković, D.; Hola, V.; Bonaventura, G.D.; Djukić, S.; Ćirković, I.; Ruzicka, F. Quantification of biofilm in microtiter plates: overview of testing conditions and practical recommendations for assessment of biofilm production by staphylococci. 2007, 115, 891–9. DOI: 10.1111/j.1600-0463.2007.apm_630.x
- Senobar Tahaei, S.A.; Stájer, A.; Barrak, I.; Ostorházi, E.; Szabó, D.; Gajdács, M. Correlation Between Biofilm-Formation and the Antibiotic Resistant Phenotype in Staphylococcus aureus Isolates: A Laboratory-Based Study in Hungary and a Review of the Literature. Infect Drug Resist. 2021, 14, 1155–68. DOI: 10.2147/IDR.S303992
- Liu, Y.; Zhang, J.; Ji, Y. Environmental factors modulate biofilm formation by Staphylococcus aureus. Sci Prog. 2020, 103, 36850419898659. doi:10.1177/0036850419898659
- The European Committee on Antimicrobial Susceptibility Testing. Breakpoint Tables for Interpretation of MICs and Zone Diameters. Version 10.0, 2020, 34–38, 84–87. Available online: https://www.eucast.org/ast_of_bacteria/previous_versions_of_documents/ (accessed on 12 February 2022).
- Reisner, A.; Krogfelt, K.A.; Klein, B.M.; Zechner, E.L.; Molin, S. In Vitro Biofilm Formation of Commensal and Pathogenic Escherichia coli Strains: Impact of Environmental and Genetic Factors. J Bacteriol. 2006, 188, 3572–81. DOI: 10.1128/JB.188.10.3572-3581.2006

Reviewer 2 Report
In this study the authors evaluated Staphylococcus aureus Colonization in Adult Patients Undergoing Tonsillectomy for Recurrent Tonsillitis.
The manuscript needs the following major improvements and this reviewer cannot recommend the manuscript before the authors performed these suggestions:
i) The introduction and discussion sections are too short
ii) The results section only contains one table and one figure;
iii) The authors only evaluated the Biofilm production capability of 24 isolates, also they did not preformed any replicated.
iv) Where is the antibiogram data?
v) The manuscript does not contain other methods to caracterize biofilm production such as crystal violet, alamarBlue and/or live dead assay.
Author Response
Thank you for valuable and constructive criticism. We addressed Your comments and suggestions point by point. Changes have been marked by green color.
- i) The introduction and discussion sections are too short
Response: Thank you! The introduction and discussion sections has been rewritten.
Introduction
The ecological niche of Staphylococcus aureus (S.aureus) in humans is the anterior nares, however it can be frequently isolated from throat, palatine tonsils and skin [1]. Approximately 20–25% of the healthy adult population have become persistently colonized by S.aureus, 60% intermittently, and 20–30% of population are non-carriers [2]. Hanson et al. (2018) reported that in the USA from 177 adults 6.2% carried S. aureus only in the anterior nares, 18.6% only in the oropharynx, and 19.8% in both sites [3]. They found out that the prevalence of oropharyngeal carriage was higher in the urban (47.3%) than rural (27.9%) environment [3]. Chmielowiec-Korzeniowska et al. (2020) study demonstrated that every third adult (32%) in Poland was an asymptomatic carrier of S.aureus, and S.aureus was recovered from the pharynx more often than from the nose or skin [2].
Previous studies have shown that there is a strong causal connection between S.aureus nasal carriage and increased risk of nosocomial infection in dialysis patients and in those undergoing surgery [1,4]. S.aureus is an etiological factor of such purulent infections as furuncles, abscesses, osteomyelitis, and sepsis. However, the role of this pathogen in the etiology of tonsillar inflammation is still unclear and complicated due to frequent carriage of S.aureus. Aside from S.aureus, also other aerobic and anaerobic pathogens are implicated in tonsil infections, for example, alpha and beta hemolytic Streptococcus (group A, C), Hemophilus influenzae, Haemophilus parainfluenzae, Enterococcus spp., Klebsiella pneumoniae, Corynebacterium spp., Peptostreptococci, Fusobacteria, Bacteroides and Veillonella [5–7]. Regardless of the widespread use of antibiotics to treat infected tonsils, episodes of tonsillitis tend to recur and form a condition known as recurrent tonsillitis (RT). Episodes of tonsillitis decrease quality of life are financial burden due to absences in school or work and health care costs [6]. RT can be treated with tonsillectomy in those patients in whom it occurs at least five or more attack episodes in a year [5,8]. If less than three episodes are observed up to the time of first presentation, the indication for surgery cannot be made until at least six episodes are achieved within the observation period [8].
Importantly, S.aureus strains can acquire broad antibacterial resistance. A part of S.aureus strains, isolated from patients with RT, were considered multidrug resistant and methicillin resistant (MRSA) [9,10]. In the Netherlands healthcare workers who are MRSA carriers achieved successful MRSA eradication only after tonsillectomy [11]. Antibiotic resistance can be explained by inadequate penetration of antibiotics into the tonsillar core, the protection of bacteria by being within epithelial cells and macrophage-like cells, the resistance of strains to the typical antibiotic treatments due to repeated antibiotic courses, the prevalence of biofilm-producing bacteria (5,6). According to Brook and Foote`s (2006) study, S.aureus strains are more often found in the tonsillar core than on the tonsillar surface [12]. S.aureus persistence in tonsillar tissues is still matter of discussion and requires further research [13].
S.aureus ability to produce multi-layered, mature biofilm may contribute to the survival of S.aureus in tonsillar tissue and play an important role in persistence of chronic infection [14–16]. Growth in biofilm provides a defence against host immune responses, can impede the access of macrophages, and increase the tolerance to antibiotics [14,15]. Biofilm-associated antibiotic tolerance is a transient state in which normally susceptible bacteria enter homeostasis that decreases sensitivity [17]. When these cells disperse and re-enter a plankton state, they regain normal antibiotic sensitivity [17].
Objectives: To evaluate the frequency and the clinical importance of S.aureus colonization and biofilm production ability in patients with RT using patient samples from tonsillar crypts during tonsillectomy, but from the throat, nasal cavity, and armpits - after tonsillectomy.
Discussion
The anterior nares represent the dominant ecological niche, other sites that can be colonized include the axilla, perineum and pharynx [4]. Elimination of nasal carriage by topical antibiotics generally leads to loss of carriage in these areas [18]. S.aureus readily recolonizes the nose, the throat and other sites within several months after antibiotic treatment [19,20]. In S.aureus carriers, infection rates are higher than in non-carriers, and patients are usually infected by the same strains with which they are colonized [17]. In our study, infected palatine tonsils were removed surgically and the presence of S.aureus in the pharynx, axilla and nares was assessed 1 year after tonsillectomy.
There are host and bacterial factors that can influence the carriage of S.aureus. The main factors predisposing to the staphylococcal infection development include age, the presence of chronic diseases and/or immunodeficiency, genetic predisposition, direct contact with healthcare settings, and hospitalization [2]. Patients enrolled in the study were young individuals (mean age 29 years) without high carriage rate host factors like HIV infection, insulin-dependent diabetes, continuous ambulatory peritoneal dialysis and haemodialysis, intravenous drug use [4].
Biofilm formation is one of the bacterial factors and is distinctive for adhesive phenotype of bacteria [4]. Bacteria in biofilm state present differential metabolic and physiological functions often rendering them more virulent and resistant to antibiotics [21]. Neopane and co-authors (2018) show that biofilm-producing S.aureus, isolated from wounds, was more resistant to various antimicrobials than the biofilm non-producers [22].
A broad range of assays for biofilm quantification in microtiter plates have been described [23,24]. We used the crystal violet assay adapted from Stepanovic et al. (2007) because it is reliable, cost-effective, straightforward and is commonly used for the quantification of biofilm production by staphylococci [25]. It is also important that this method can be easily performed by other investigators. Because both living and dead, as well as matrix are stained by crystal violet, it is poorly suited for differentiation between living and dead cells and susceptibility testing of biofilms cannot be done [23]. The drawbacks of the method do not affect the research question being investigated in our study. However other methods as flow cell systems would increase data reliability and would let more detailed biofilm investigation.
In our study, emphasis was put on biofilm forming ability of S.aureus isolated strains. S.aureus strains, isolated from the tonsillar crypts, as compared to isolates collected from other body sites, had greater capacity to produce biofilms. Biofilm producing S.aureus strains were mostly isolated from tonsillar crypts and were susceptible to majority of the tested antibiotics. Only one isolate was identified as MRSA, it showed wider spectrum of resistance and was a weak biofilm producer. The isolate was obtained from 25 years old female without co-morbidities, with 5 episodes of tonsillitis per year for the last 3 years. After tonsillectomy MRSA strain was eradicated. MRSA strains are detected in RT patients in other studies also. In the study conducted by Katkowska et al., MRSA strain was isolated from the tonsils in one out of 118 adult patients, in two out of 73 children qualified for tonsillectomy in Poland [7,10]. The role of biofilm formation and antimicrobial resistance of MRSA and methicillin-susceptible S.aureus (MSSA) is unclear [26]. The environmental factors (temperature, pH, glucose level, type of media and others) influence bacterial biofilm production, therefore these factors should be accounted in biofilm research. To compare results from different studies one should use similar or even the same biofilm method and environmental factors [27]. Our study results showed that in 10 patients tonsillectomy resulted in no growth of S.aureus strains a year later, therefore tonsillectomy could prevent bacterial colonization in one-year period. However, to prove such phenomena it should be investigated in bigger cohorts.
Our study underlines the immense importance of studying S.aureus colonization to understand the pathology of staphylococcal disease. Current efforts to interrupt carriage rely on the use of antibiotics, but the development of efficacious anti-biofilm S. aureus therapies is a new and necessary perspective.
- ii) The results section only contains one table and one figure;
Response: Per Your recommendation additional table in the results section was added. Please see the attachment.
Table 2. Antibiotic resistance among S. aureus isolated from patients with RT. FOX, cefoxitin; CRO, ceftriaxone; P, benzylpenicillin; AMP, ampicillin; AMS, ampicillin-sulbactam; AUG, amoxicillin-clavulanic acid; CIP, ciprofloxacin; AK, amikacin; E, erythromycin; CD, clindamycin; C, chloramphenicol. S, sensitive; R, resistant; I, intermediate.
iii) The authors only evaluated the Biofilm production capability of 24 isolates, also they did not preformed any replicated.
Response: Thank you! We did evaluate biofilm formation from all S.aureus isolates, in case we would have isolated more of them biofilm evaluation would also be performed for them. For our biofilm formation assay we used 8 wells per strain, each experiment was triplicated. You can find this in the section “Materials and Methods”, page 7, lines 216 - 218. To make reliable and comparable results for each plate we used the negative control.
- iv) Where is the antibiogram data?
Response: Thank you, this is corrected now. Please see the attachment.
Table 2. Antibiotic resistance among S. aureus isolated from patients with RT. FOX, cefoxitin; CRO, ceftriaxone; P, benzylpenicillin; AMP, ampicillin; AMS, ampicillin-sulbactam; AUG, amoxicillin-clavulanic acid; CIP, ciprofloxacin; AK, amikacin; E, erythromycin; CD, clindamycin; C, chloramphenicol. S, sensitive; R, resistant; I, intermediate.
- v) The manuscript does not contain other methods to characterize biofilm production such as crystal violet, alamarBlue and/or live dead assay.
Response: Thank you! For our study we used crystal violet assay adapted by Stepanovic et al. (2007). Per Your recommendations explanations were added in the “Discussion”, page 5-6, lines 143-152, and the “Materials and Methods”, page 7, lines 209-211.
- Discussion
A broad range of assays for biofilm quantification in microtiter plates have been described [23,24]. We used the crystal violet assay adapted from Stepanovic et al. (2007) because it is reliable, cost-effective, straightforward and is commonly used for the quantification of biofilm production by staphylococci [25]. It is also important that this method can be easily performed by other investigators. Because both living and dead, as well as matrix are stained by crystal violet, it is poorly suited for differentiation between living and dead cells and susceptibility testing of biofilms cannot be done [23]. The drawbacks of the method do not affect the research question being investigated in our study. However other methods as flow cell systems would increase data reliability and would let more detailed biofilm investigation.
- Materials and Methods
5.2. Biofilm Growth Using Cristal Violet Assay
Crystal violet assay adapted from Stepanovic et al. (2007) was used for the in vitro cultivation and quantification of bacterial biofilms [25].

Round 2
Reviewer 2 Report
The authors did some alterations to their manuscript and this is now a more stronger study.
However, the level of discussion is somehow incipient, the manuscript contains only 29 references.
Also, the manuscript still lacks more results. The characterization of biofilms at least for some isolates, in this reviewer opinion, needs few more methods: confocal microscopy and/or SEM.
Round 3
Reviewer 2 Report
The authors improved their manuscript, but this reviewer thinks that the authors did not altered significantly the quality of results section.
Figure 1- I strongly recommend that this should be improved. It does not have the adquated quality for a publication. Also, the authors should try to do an effoirt to associete the data dipited in this figure with the main text (including results depicted in table 1 and figure 2).
